# Human Resource Professionals' Responses to Workplace Bullying

**Kelly Rae and Annabelle M. Neall ***

School of Psychology, University of Queensland, Brisbane, QLD 4072, Australia
* Correspondence: a.neall@uq.edu.au

**Abstract:** Workplace bullying manifests in significant costs to individuals and organisations. The obligation to resolve such cases largely falls on Human Resource Professionals (HRPs). Little is known, however, about the antecedents to HRPs' helping behaviour in these scenarios. Using the attribution–emotion model of stigmatisation, this study explored how HRPs are influenced in their response to workplace bullying. Australian HRPs ($n$ = 84) were assigned to one of four experimental vignette scenarios, differing in target (approach/avoidance coping) and perpetrator (effort vs. non-effortful response) behaviour. The results revealed that targets who fail to act to resolve situations of bullying were regarded as more responsible and less likely to receive help, but HRPs were more sympathetic and inclined to help non-responsive perpetrators when the targets also avoided the situation. The findings indicate two key areas for training and development that could improve HRPs evaluations and management of workplace bullying.

**Keywords:** workplace bullying; human resources; intervention; bystander

## 1. Introduction

Systematic forms of aggression and social exclusion occurring persistently and over a period of time, brought about by superiors, co-workers, and subordinates—otherwise known as workplace bullying—are a serious and prevalent problem within organisations [1,2]. Taking form as verbal and non-verbal attacks, threats interfering with another's work, social isolation, personal attacks, insulting remarks, gossip, and humiliation [3], bullying behaviours manifest in various health consequences for targets, including stress, depression, anxiety and psychosomatic symptoms [4]. Beyond this, impacts to organisations include decreased morale, increased absenteeism, employee intention to leave, and turnover [5]. Financially, workplace bullying places considerable strain on organisations, consisting of workers compensation claims and lost productivity due to depression, psychological stress [3], sick days, and decreased morale [6,7].

The social, financial, and organisational consequences of workplace bullying have driven considerable research into how to prevent and minimise its occurrence. Reviews of workplace bullying interventions have continually ascertained that multi-level approaches are most effective is preventing and protecting workers [8–10], and that anti-bullying policies and training are the most common tactical approaches [11]. The best practice in the implementation of these approaches includes organisational leader-led communication and enforcement [11–13]; in reality, however, implementation of policy through leaders is not always effective [14]. Leader cohorts are known to be reluctant to take up their role [15], inconsistent in application [14], or lacking in the ability to execute challenges amongst cultures [16]. As a result, the challenge is repeatedly put to Human Resource Professionals (HRPs) to implement policy, intervene in the case of poor behaviour [17–19] and provide a psychologically safe, fair, and just workplace, in line with workplace health and safety legislation [20]. Meanwhile, the HRP role has morphed over time and now must protect and represent the needs of the organisation as an entity, and management and employees [21–24]. Mokgolo and Barnard [25] documented the experiences of HRPs who deal with workplace bullying, noting paradoxical role demands, along with a lack

of support, and a lack of decision-making power, as the key challenges faced. A similar study of HRPs by Cowan [26] revealed discrepancies between prescribed policy versus real-life management of such cases, causing confusion for HRPs and inequitable outcomes for targets. Compounding the issue, HRPs have been known to demonstrate bias or stigmatization towards employees [27,28]. A review of 65 studies found disciplinary decisions were made based on HRPs internal attributions of poor performance rather than actual performance [27], and decisions changed based on whether the reason for a problem was perceived to be internal or external to a person [29]. Stigmatization was demonstrated toward those with medical conditions [28] and insecurely tenured backgrounds (i.e., previously laid-off workers) [30]. Additionally, HRPs were found to raise "symbolic violence" against those who made bullying claims against a manager, whereby the manager was cleared, and the organization protected at the expense of the employee [31]. Ostensibly, HRPs are left in a position where they may not be able to adequately deal with workplace bullying due to the breadth of expectations and the competing tensions of their roles.

While no studies to date have explored whether HRPs specifically stigmatise workers who have been bullied, a parallel stream in the literature on bystander intervention suggests that, when targets of bullying are perceived as responsible for their situation, they are less likely to receive help from others [32]. Specifically, Mulder et al. [32] found that targets of bullying who appeared to avoid the situation as a coping response elicited less pity for the situation and subsequently less help from bystanders in the workplace. It is pertinent to note, however, that typical bystanders usually do not have an obligation (other than a moral one) to intervene in the face of workplace wrongdoing [22] and may not be aware of the full details of the situation (i.e., attributions may be based on piecemeal information trickling through multiple sources or based on preestablished loyalties or preferences for the target or perpetrator). In contrast, HRPs, by virtue of their role, are required to thoroughly investigate and take an objective view of the situation, thus lessening the likelihood (in theory) of incorrectly attributing responsibility and blame to the incorrect party.

There is limited research focussed on the antecedents of HRPs' workplace behaviour bullying scenarios, despite the pivotal role they are required to play, nor whether perceptions of the perpetrator's behaviour influence their willingness to help. If HRPs are prone to the same cognitive bias in restricting intervention to workers not deemed as 'responsible' for their plight, one may also raise concerns regarding the capacity of organisations to effectively deal with and mitigate the costs of workplace bullying. Establishing whether HRPs make attributions that affect the outcome of their investigations adds to the crucial literature base on helping behaviour as it relates to bullying. To address this gap and to support best practice recommendations for human resource management, this study sought to explore HRP responses to workplace bullying from the perspective of the attribution–emotion theory of stigmatization. Using experimental vignette methodology, the project investigated whether HRPs are influenced in their response to workplace bullying by the behaviours exhibited by the target and/or the perpetrator of the bullying.

*1.1. Helping Behaviour*

The beginnings of scholarly research on helping behaviour are most often attributed to the death of Kitty Genovese in 1964, after it became apparent that 39 of her building cohabitants witnessed (to some degree) her murder but did not intervene or seek help from the police. Those that were aware of the situation and did not act were thought to have been immobilized by what would become known as the 'bystander effect' [33]. In a series of studies, Darley and Latane [34] found that the greater the number of bystanders present at an emergency, the less likely any one individual would take action; that is, responsibility for taking action is diffused among those present, diluting the need for (and associated consequences) taking action in the situation. Piliavin, Rodin, and Piliavin [35] sought to replicate this effect in a series of field settings, testing the responses of bystanders riding the New York City subways. They found that victims requiring assistance because they were deemed ill (and thus not responsible for their plight) were more likely to elicit help than

victims who appeared intoxicated (i.e., were responsible for the situation). Further, their study did not find support for the theory of diffused responsibility, indicating that there was more to helping behaviour motivations than the number of other bystanders present.

More recent investigations of helping behaviour have drawn on the attribution–emotion model of stigmatization as an underpinning framework [36,37]. The theory posits that the behaviours of a stigmatized person influence the judgements about or attributions of that person, and subsequent intentions to aid or help that person [36,38]. That is, judgements made about the perceived responsibility a victim holds, in turn, trigger a positive or negative affective (i.e., emotional) response, which consequently affects whether the judger intends to help or not [38]. Studies situated in academic and social settings have attempted to understand how helping behaviour is elicited according to situational causality (i.e., internal, or external locus; temporary or permanent stability) and controllability. In the first study, helping behaviour was operationalized as lending notes to a classmate who needed assistance because he did not try to take notes (internal, controllable) or could not physically take notes (internal, non-controllable), because the professor (external) was or was not able to deliver a clear lecture. Helping behaviour was only limited in the context of an internal cause and controllable circumstances, that is, when the person needing help was deemed responsible for and in control of their plight. A second study presented a common social interaction: someone needing help on the train. When asked to rate the cause of the event and associated emotions evoked by the situation, participants indicated negative affect toward an inebriated person and tended more sympathy and concern for an ill (fainting) person; that is, less care was provided for victims personally (internal locus) responsible (controllable) for their situation. Similarly, a student who made no effort to achieve high marks at university (controllable) elicited greater anger than a student who was physically disadvantaged (disabled), while the latter provoked greater pity and sympathy [39]. Collectively, the studies demonstrated that attributions of responsibility are mediated by positive and negative emotional responses, and that helping behaviour is most likely to occur when someone is deemed not responsible for his or her predicament.

## 1.2. Responsibility and Coping Behaviours

Brickman et al. [40] posits that the label 'help' carries the implication that a person is not responsible for solving their own problem, and that this in itself is a dilemma for some situations. They identified two types of responsibility: one related to cause (onset of the problem) and the other to solution (offset of the problem). Karasawa [41] examined the variables of onset and offset responsibility and discovered that regardless of the onset responsibility attributed (i.e., illness), participants held the expectation that effort would be applied to rectify the situation. A lack of effort to change the situation was subsequently associated with more anger, less pity, and less helping intentions. Relative to situations of workplace bullying, it reasons that regardless of how bullying was instigated, targets of bullying may still be perceived as responsible for the continuation of the bullying if they do not take active steps to amend the situation, which, in turn, may affect whether help is directed toward them.

Whether someone is attempting to handle their situation [coping] is known to influence whether they are perceived to be responsible, and influences whether others will help them [40,42]. Further, where someone is deemed not to be coping well, a perceiver of their behaviour is more likely to experience anger toward them, have less sympathy, and subsequently not be willing to help them [43]. Coping behaviour is often dichotomised as approach-oriented: taking action to confront the problem; or avoidance oriented: taking no action or acting in a way that avoids the problem [44]. Mulder and colleagues [32,37,45] tested this idea in the context of workplace bullying. The authors found that bystanders evaluated victims who exhibited approach-oriented behaviour to be more self-reliant and less responsible for the continuation of the bullying, which also increased their sympathy for and helping intentions toward the target. While these studies support the argument that bystanders (i.e., co-workers) make responsibility attributions about situations of workplace

bullying relative to behaviours exhibited by the target, it also stands to reason that HRPs may make similar attributions about displayed behaviours and their relationship to offset responsibility regarding workplace bullying.

*1.3. Current Study*

Leveraging the coping literature and the attribution–emotion model of stigmatization, the current study examined the effect of target and perpetrator behaviours on HRPs' attributions, and subsequent intentions to intervene in cases of workplace bullying. Using experimental vignette methodology, we sought to understand whether HRPs attribute perceived responsibility for the continuation of the bullying to either the target or perpetrator, based on their behaviours. A secondary aim sought to understand HRPs experiences of emotions, such as anger or sympathy, in situations of workplace bullying and whether such emotions mediate willingness to help.

Coping types were manipulated to ascertain whether avoidance-coping behaviour (i.e., avoiding the bully and the bullying situation) or approach-coping behaviour (i.e., confronting the bully and trying to stop the bullying situation), elicited stronger perceptions of anger, sympathy, and greater helping behaviour towards the target of bullying. This study's hypotheses were the following:

**Hypothesis 1a.** *Targets who exhibit approach-coping behaviour will be evaluated as less responsible for the continuation of the bullying than targets who exhibit avoidance-coping behaviour.*

**Hypothesis 1b.** *Targets who exhibit approach-coping behaviour will evoke less anger and more sympathy than targets who exhibit avoidance-coping behaviour.*

**Hypothesis 1c.** *Targets who exhibit approach-coping behaviour will be evaluated as more deserving of help than targets who exhibit avoidance-coping behaviour.*

Response behaviour was operationalised as effort/no-effort behaviour, whereby perpetrators respond to feedback about their bullying behaviour and resolve to cease such behaviour but do not follow through (no effort), or perpetrators respond to feedback and amend their behaviour (effort). The following was expected:

**Hypothesis 2a.** *Perpetrators who exhibit effort behaviour will be evaluated as less responsible for the continuation of the bullying than perpetrators who exhibit no-effort behaviour.*

**Hypothesis 2b.** *Perpetrators who exhibit effort behaviour will evoke less anger and more sympathy than perpetrators who exhibit no-effort behaviour.*

**Hypothesis 2c.** *Perpetrators who exhibit effort behaviour will be evaluated as more deserving of help than perpetrators who exhibit no-effort behaviour.*

## 2. Materials and Methods

*2.1. Design, Participants and Procedure*

The study utilised a 2 (coping behaviour: approach; avoidance) x 2 (response behaviour: effort; no effort) randomized between-persons design, using experimental vignette methodology. Participants were Australian adults, aged over 18 years of age and currently or recently employed as Human Resources Professionals ($n$ = 84). The majority of participants were female (82%), and the mean age was 39.2 years ($SD$ = 11.94). The most common role title was HR Manager, followed by HR Officer, HR Advisor, and HR Partner.

Data were collected following university HREC ethics approval. Participants were recruited via professional (i.e., LinkedIn) and personal (i.e., Facebook) networks, and a Qualtrics paid panel, who recruited participants who met the criteria above (i.e., HRPs who were currently or within the previous 12 months employed in a paid HRP role, aged

18 years and over, and based in Australia). The survey yielded 158 unique responses. Thirty-two respondents did not proceed beyond the screening questions (i.e., two participants were aged under 18 years, and 30 did not meet the criteria of holding a current or previous HR role). A further 29 responses were incomplete and thus removed from the dataset. Thus, the final sample comprised 84 responses.

Participants were randomly assigned to one of four workplace bullying scenarios (i.e., approach versus avoid coping behaviour and effort versus no-effort response behaviour). Following the information and consent page, participants were asked to vividly imagine the scenario as having occurred in front of them, in their current or former workplace. They were then asked to consider how they, in their role as a Human Resource Professional, would respond to the situation of bullying. After presentation of the vignette, participants were asked to rate the level of avoidance shown by the target in the bullying scenario (i.e., a manipulation check of the IV) and the level of behavioural change demonstrated by the perpetrator (i.e., manipulation check of the IV). Participants then responded to a series of questions pertaining to the dependent variables (i.e., responsibility for the continuation of the bullying, level of sympathy for the target/perpetrator, level of anger toward the target/perpetrator, and intention to help the target/perpetrator).

### 2.2. Measures

*Vignette content.* Drawing on best practice recommendations [46–48], the vignettes were designed specifically for this study. Each vignette began with a statement describing a period of workplace bullying (i.e., negative behaviours occurring over 6 months and escalating over time, where one party is inferior to the other), aligning with the Australian legislative definition of bullying [7]. The bullying was said to occur between two co-workers with differing levels of organisational power; specifically, the perpetrator was reported as holding a longer period of tenure with the organisation compared to the target, a manifestation of expert, referent, or informational power [49]. See Appendix A for the full scenario. Depending on the experimental condition, a statement describing the target's coping behaviour (i.e., whether they appeared to approach or avoid the perpetrator) and a subsequent statement regarding the perpetrator's response behaviour (i.e., whether they responded to feedback or continued to act in the same way toward the target) were presented. The two avoidance-coping conditions were longer (157 and 160 words) than the two approach-coping conditions (122 and 125 words), as they had to introduce a third party that reported the issue.

The text for coping conditions was drawn from Mulder et al. [32], where bystanders were identified as more likely to attribute responsibility for the continuation of the bullying if the target exhibited avoidance coping behaviour. 'Avoidance' is defined as actively moving away from the situation, whereas 'Approach' is defined as taking active steps to address the situation. This was operationalized as the target directly asking the perpetrator to stop harassing them (approach) or the target deeming themselves incapable of approaching the perpetrator, and moving away from the workspace (avoidance). The constructs of 'Effort' and 'No Effort' were conceptualised from Karasawa [41] and adapted to statements for workplace bullying. Specifically, perpetrator effort behaviour was operationalized as the perpetrator resolving to do things differently and reducing their harmful behaviours, while perpetrator no-effort behaviour was depicted as hearing the feedback and continuing to act in the same way.

The vignettes were validated for content and face validity in a separate study of 38 Australian employees. There was a significant main effect of coping behaviour on ratings of the target's behaviour; $F(1, 36) = 129.35$, $p < 0.001$, partial $\eta^2 = 0.78$, as participants rated the target with the avoidance-coping behaviour as more avoidant ($M = 4.32$, $SE = 0.16$) than the target with approach-coping behaviour ($M = 1.91$, $SE = 0.16$), showing a mean difference of 2.42, 95% CI [1.99, 2.85], $p < 0.001$. Similarly, there was a significant main effect of response behaviour on ratings of the perpetrator's behaviour, $F(1, 36) = 82.70$, $p < 0.001$, partial $\eta^2 = 0.70$; specifically, participants rated the perpetrator exercising effort as more

effortful (*M* = 4.27, *SE* = 0.15) than the no-effort behaviour condition (*M* = 1.93, *SE* = 0.26). Thus, the vignettes were deemed valid for the study aims.

*Manipulation check.* The effectiveness of the manipulation was measured by two questions against a 6-point Likert scale: "To what extent was [*target*] avoidant of the problems he was facing in this scenario?" and "How effortful was [*perpetrator*] in modifying his behaviour in this scenario?".

*Dependent variables.* All variables of interest were presented on a 7-point Likert scale (*1 = not at all; 7 = very much*). Cognitive, emotional, and behavioural variables were measured in sequence.

*Cognition.* Perceived responsibility was measured with two items: "Is [*target/perpetrator*] responsible for the continuation of the negative treatment by [*perpetrator*]?" and "Does the way [*target/perpetrator*] behave in the situation contribute to its continuation?". These items were adapted from Mulder et al. [32], who explored bystander cognition in the face of workplace bullying.

*Emotional responses.* Two emotions were measured based on items used by Struthers, Weiner, and Allred [50] and Mulder et al. [32]: sympathy and anger.

*Sympathy.* Sympathy was measured with three items questioning the level of sympathy, pity, and compassion towards the target/perpetrator. An example question was "To what extent do you have sympathy for [*target/perpetrator*]?".

*Anger.* Anger was measured with three items: how annoyed, upset, and angry the respondent felt about the target and perpetrator. An example question was "To what extent are you annoyed by [*target/perpetrator*]'s behaviour?".

*Helping intention.* Helping intention was measured with three items designed to test the confidence, willingness, and probability of aiding the target/perpetrator. Questions were again adapted from Mulder et al. [32]. An example question was "How willing would you be to help [*target/perpetrator*]?"

### 2.3. Analyses

All data were cleaned, and assumption testing was conducted. Independent t-tests were run to assess the effectiveness of the manipulation (independent) variables. Hypotheses 1a–c and 2a–c were tested via separate MANOVAs, grouping outcome variables by target or perpetrator.

## 3. Results

### 3.1. Manipulation Checks

To ensure a meaningful difference between the approach-coping behaviour and avoidance-coping behaviour scenarios, an Independent Samples t-test was conducted. The results indicated that, overall, participants rated the target as exhibiting more avoidance behaviour in the avoidance-coping behaviour scenario (*M* = 4.30, *SD* = 1.74), compared to the approach-coping behaviour scenario (*M* = 2.80, *SD* = 1.17); a statistically significant difference of 1.50, $t(82) = 4.60$, $p < 0.001$, 95% CI [0.85, 2.16], *d* = 1.02. Thus, the manipulation of coping condition was deemed successful.

A second Independent Samples t-test was run to determine if there was a difference in ratings of effort between the effortful-response scenario and no-effort response scenario. Effortful-response behaviour (*M* = 3.54, *SD* = 1.47) was perceived as more effortful than no-effort response behaviour (*M* = 2.72, *SD* = 2.02), a statistically significant difference of 0.82, $t(82) = -2.12$, $p < 0.05$, 95% CI [−1.58, −0.05], *d* = 3.54. Accordingly, the manipulation of the response condition was considered successful.

### 3.2. The Effect of Coping Behaviour on Perceptions of the Target

To assess the effect of coping behaviour and response behaviour on four dependent variables (i.e., perception of continuation of responsibility, sympathy towards the target, anger felt toward the target and indication of helping intention for the target), a two-way MANOVA was conducted. There was a statistically significant main effect of coping

behaviour on the dependent variables, $F(4, 77) = 12.50$, $p < 0.001$, Pillai's Trace = 0.39, partial $\eta^2 = 0.39$, and a main effect of response behaviour on the dependent variables $F(4, 77) = 6.15$, $p < 0.001$, Pillai's Trace = 0.24, partial $\eta^2 = 0.24$, but the interaction was not significant.

*Main effects.* Follow up univariate main effects tests demonstrated that there was a statistically significant main effect of coping behaviour on perceived responsibility, anger and helping but not for sympathy (see Table 1). Bonferroni post hoc analyses showed that HRPs attributed more responsibility to the target when they displayed avoidance-coping behaviour ($M = 6.85$, $SD = 3.78$), compared to approach-coping behaviour ($M = 4.18$, $SD = 2.56$); this demonstrated a mean difference of 2.71, 95% CI [1.33, 4.09], $p < 0.001$, thus supporting Hypothesis 1a. Similarly, HRPs were angrier at the target when they employed avoidance-coping behaviour ($M = 10.67$, $SD = 5.29$), compared to approach-coping behaviour ($M = 5.22$, $SD = 3.80$), a mean difference of 5.60, 95% CI [3.69, 7.50], $p < 0.001$. However, as coping behaviour did not have a significant effect on sympathy felt for the target, Hypothesis 1b was partially supported. Finally, targets were less likely to be helped by HRPs when they responded with avoidance-coping behaviour ($M = 15.23$, $SD = 5.16$), when compared to approach-coping behaviour ($M = 17.75$, $SD = 3.58$), a mean difference of 2.74, 95% CI [−4.47, −1.01], $p < 0.01$, thereby providing support for Hypothesis 1c.

**Table 1.** Effects of Target Coping Behaviour and Perpetrator Response Behaviour on Perceptions of the Target.

| Independent Variable | Dependent Variable | df | F | Sig. | Partial Eta Squared | Observed Power |
|---|---|---|---|---|---|---|
| Coping behaviour | Responsibility | 1 | 153.04 | 0.000 | 0.160 | 15.23 |
| | Sympathy | 1 | 0.07 | 0.952 | 0.000 | 0.00 |
| | Anger | 1 | 653.39 | 0.000 | 0.299 | 34.11 |
| | Helping | 1 | 156.02 | 0.002 | 0.110 | 9.89 |
| Response behaviour | Responsibility | 1 | 1.62 | 0.207 | 0.020 | 0.24 |
| | Sympathy | 1 | 3.86 | 0.053 | 0.046 | 0.49 |
| | Anger | 1 | 7.32 | 0.008 | 0.084 | 0.76 |
| | Helping | 1 | 16.61 | 0.000 | 0.172 | 0.98 |

*Note*: MANOVA: IVs: Target (i.e., approach or avoid) and Perpetrator (effort or non-effort) response behaviour.

Follow up univariate main effects were conducted for the effect of response behaviour on target responsibility, sympathy, anger and helping intention. There was an unexpected significant main effect for anger and helping but not for responsibility or sympathy (see Table 2). Bonferroni post hoc analyses showed that HRPs felt less anger toward the target when the perpetrator made an effort to change their behaviour ($M = 6.73$, $SD = 4.68$), compared to no-effort response behaviour ($M = 8.86$, $SD = 5.70$); the mean difference was 2.59, 95% CI [−4.50, −0.69], $p < 0.01$. Additionally, HRPs had greater helping intentions toward the target when the perpetrator amended their behaviour ($M = 18.22$, $SD = 3.06$), versus when no effort was made to amend their behaviour ($M = 14.95$, $SD = 5.17$); the mean difference was 3.55, 95% CI [1.81, 5.28], $p < 0.001$.

**Table 2.** Effects of Target Coping Behaviour and Perpetrator Response Behaviour on Perceptions of the Perpetrator.

| Independent Variable | Dependent Variable | df | F | Sig. | Partial Eta Squared | Observed Power |
|---|---|---|---|---|---|---|
| Response behaviour | Responsibility | 1 | 11.30 | 0.001 | 0.124 | 0.913 |
| | Sympathy | 1 | 11.90 | 0.001 | 0.130 | 0.926 |
| | Anger | 1 | 8.85 | 0.004 | 0.100 | 0.836 |
| | Helping | 1 | 2.13 | 0.149 | 0.026 | 0.302 |

**Table 2.** *Cont.*

| Independent Variable | Dependent Variable | df | F | Sig. | Partial Eta Squared | Observed Power |
|---|---|---|---|---|---|---|
| Coping behaviour | Responsibility | 1 | 7.79 | 0.007 | 0.089 | 0.787 |
| | Sympathy | 1 | 12.38 | 0.001 | 0.134 | 0.935 |
| | Anger | 1 | 0.00 | 0.947 | 0.000 | 0.050 |
| | Helping | 1 | 0.15 | 0.695 | 0.002 | 0.067 |
| Response x coping behaviour | Responsibility | 1 | 4.64 | 0.034 | 0.055 | 0.567 |
| | Sympathy | 1 | 5.68 | 0.020 | 0.066 | 0.653 |
| | Anger | 1 | 2.98 | 0.088 | 0.088 | 0.399 |
| | Helping | 1 | 8.70 | 0.571 | 0.004 | 0.087 |

*Note*: MANOVA: IVs: Target (i.e., approach or avoid) and Perpetrator (effort or non-effort) response behaviour.

### 3.3. The Effect of Response Behaviour on Attributions of the Perpetrator

A two-way MANOVA assessed the impact of coping and response behaviours on four dependent variables: perceived responsibility, sympathy for the perpetrator, anger felt toward the perpetrator and helping intention for the perpetrator. A significant main effect was obtained for response behaviour on the dependent perpetrator variables $F_{(4, 77)} = 7.51$, $p < 0.001$, Wilks' $\Lambda = 0.72$, partial $\eta^2 = 0.28$, and a significant main effect of coping behaviour on the dependent variables $F_{(4, 77)} = 4.83$, $p = 0.002$, Wilks' $\Lambda = 0.80$, partial $\eta^2 = 0.20$. Additionally, there was a significant interaction between coping behaviour and response behaviour, $F_{(4, 77)} = 2.94$, $p = 0.025$, Wilks' $\Lambda = 0.87$, partial $\eta^2 = 0.13$.

Main effect follow-up univariate tests with Bonferroni post hoc analysis for response behaviour determined that there were significant main effects on perceived responsibility, anger and helping intention toward the perpetrator, but not for sympathy (see Table 2). Contrary to expectations, more responsibility was attributed to the perpetrator when they displayed an effortful response ($M = 12.01$, $SD = 2.27$) compared to a no-effort response ($M = 10.12$, $SD = 3.57$), a statistically significant mean difference of 2.06, 95% CI [0.84, 3.29], $p = 0.001$, and in opposition to Hypothesis 2a. There was also significantly less sympathy for the perpetrator in the effortful condition ($M = 6.66$, $SD = 4.31$) than the no-effort condition ($M = 9.63$, $SD = 5.22$), with a mean difference of $-3.31$, 95% CI [$-5.22$, $-1.40$], $p = 0.001$. HRPs were significantly angrier toward the perpetrator when they were effortful ($M = 14.57$, $SD = 4.36$) than when they showed no effort ($M = 11.53$, $SD = 4.98$); the mean difference was 3.033, 95% CI [1.00, 5.06], $p < 0.001$. Thus, Hypothesis 2b was unsupported. Additionally, there was no significant difference for helping intention when the perpetrator made an effort ($M = 13.61$, $SD = 5.44$), compared to when they made no effort ($M = 11.95$, $SD = 4.82$), with a mean difference of 1.65, 95% CI [$-0.60$, 3.91], $p = 0.149$; thus, Hypothesis 2c was also unsupported.

Main effect follow up Bonferroni tests found significant effects of coping behaviour on perceptions toward the perpetrator for responsibility and sympathy, but not for anger or helping; there were no hypotheses for the effect of target behaviour on perpetrator perceptions. More responsibility was ascribed to the perpetrator when the target displayed approach-coping behaviour ($M = 11.80$, $SD = 2.36$) than when the target showed avoidance-coping behaviour ($M = 10.20$, $SD = 3.61$), a statistically significant mean difference of 1.71, 95% CI [0.49, 2.93], $p = 0.007$. HRPs had significantly more sympathy for the perpetrator when the target exhibited avoidance-coping ($M = 9.85$, $SD = 5.47$) than approach-coping behaviour ($M = 6.66$, $SD = 4.02$), 3.37, 95% CI [1.47, 5.23], $p = 0.007$.

*Interaction Effects*. Follow up univariate and Bonferroni post hoc analyses were used to determine the effect that the interaction between coping behaviour and response behaviour had on each dependent variable. There was a statistically significant interaction effect on perceived responsibility and sympathy for the perpetrator but not for anger or helping intention toward the perpetrator (see Table 2). This indicated that both the target and

perpetrator's behaviour was taken into consideration when assessing whether the perpetrator was responsible for the continued bullying and subsequent sympathy felt toward the perpetrator.

Simple main effects were assessed to understand the effect response behaviour had on each level of coping behaviour. There was a simple main effect for perceived responsibility (see Figure 1). For the avoidance-coping behaviour, higher perpetrator effort led to more perceived responsibility for the perpetrator, $F(1, 80) = 14.57$, $p < 0.001$, partial $\eta^2 = 0.16$, a statistically significant mean difference of 3.39, 95% CI [1.62, 5.16]. For the approach-coping condition, there was no difference between the response behaviour conditions, $F(1, 80)$, $p = 0.76$, partial $\eta^2 = 0.01$. There was also a simple main effect on sympathy for the perpetrator (see Figure 2). For the avoidance-coping condition, higher perpetrator effort led to lower sympathy for the perpetrator, $F(1, 80) = 16.28$, $p < 0.001$, partial $\eta^2 = 0.17$, a statistically significant mean difference of 5.60, 95% CI [1.62, 5.16]. For the approach-coping condition, there was low sympathy toward the perpetrator, regardless of the effort response demonstrated $F(1, 80) = 0.60$, $p = 0.44$, partial $\eta^2 < 0.01$.

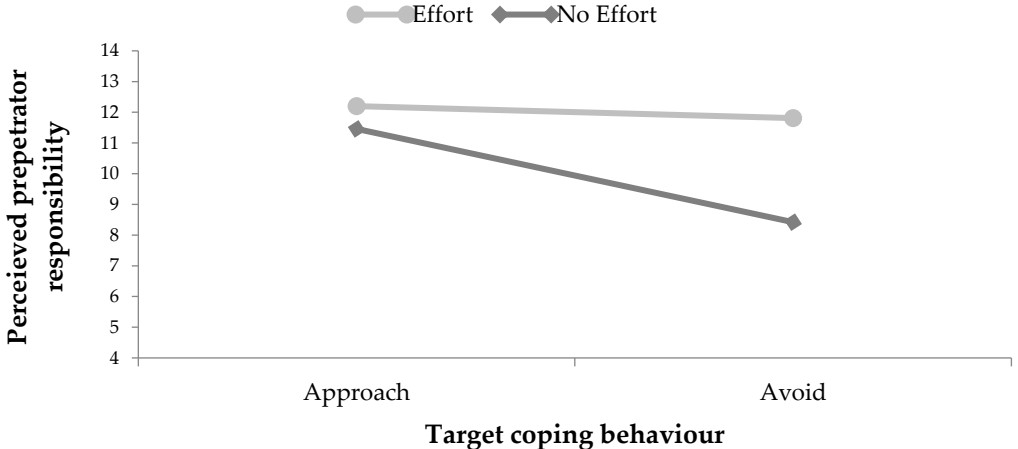

**Figure 1.** Estimated marginal means of the interaction of perpetrator response behaviour and target coping behaviour on perceived perpetrator responsibility.

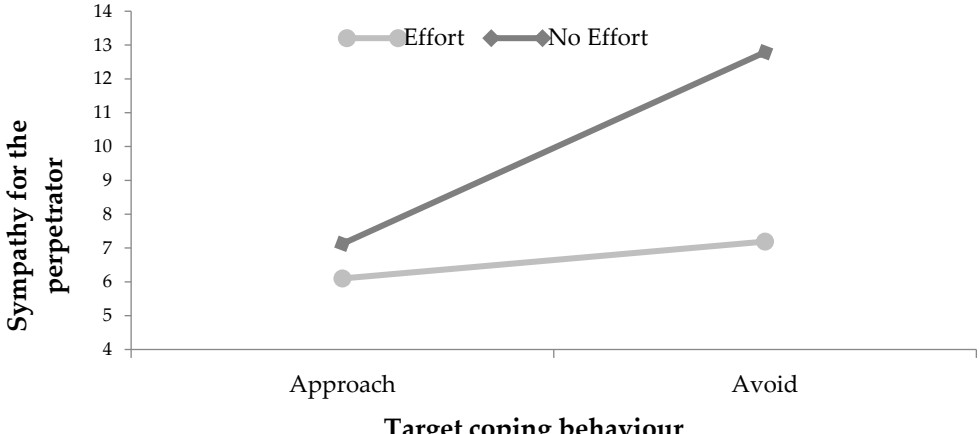

**Figure 2.** Estimated marginal means of the interaction between perpetrator response behaviour and target coping behaviour upon felt sympathy for the perpetrator.

## 4. Discussion

Workplace bullying is a problem with costly consequences for all involved [4,5]. HRPs are expected to play a role in helping to resolve such incidents [51,52], though the very nature of their roles causes tension when trying to assist both organisational leaders and employees [21,22]; this leads to bias towards certain attributes, which in turn can influence

helping behaviour (Giel et al., 2010; Karren & Sherman, 2012). To enhance an organisation's ability to effectively address workplace bullying, it is important to understand how HRPs evaluate situations of workplace bullying. To that end, this study aimed to identify how HRPs attribute responsibility to the target and/or perpetrator in a bullying scenario, and whether corresponding emotions arising from the attribution of responsibility affect subsequent intentions to help. The findings offer nuanced understandings of how HRPs appraise cases of workplace bullying and ascribe assistance, but also highlight significant disparities in our grasp of HRP decision making.

## 4.1. Advancement of Knowledge

The key contribution arising from this study lies in a new understanding of how HRPs attribute responsibility in the continuation of bullying, and the subsequent help behaviour in the context of workplace bullying. Couched in the attribution–emotion model of stigmatization, our findings offer supporting and conflicting evidence for the model, and new insights into the antecedents of helping behaviour in HRPs. As hypothesised, data from this study aligned with the attribution–emotion theory of stigmatisation, relative to workplace bullying scenarios for the target of bullying. Specifically, less responsibility (for the continuation of the situation) was attributed to targets that displayed approach-coping behaviour (i.e., those who asked the perpetrator to amend their behaviour), which in turn triggered less anger and increased helping intentions. By contrast, targets who exhibited avoidance-coping behaviour elicited attributions of greater responsibility, more anger, and lower helping intentions. Unexpectedly, HRPs reported high sympathy regardless of the target's coping behaviour style. With the exception of felt sympathy, these findings align with previous studies of bystander intervention [32,45] and the attribution–emotion model of stigmatization [36,53]. In many emergency situations, helping behaviour is contingent on the bystander's attributions of the target and the situation at hand.

However, while targets were judged for their actions alone, a higher level of complexity was involved in HRPs attributions of the perpetrator. Although we predicted that effortful behaviour (i.e., taking steps to alter behaviour after being made aware of their bullying) would lead to less responsibility, less anger, more sympathy, and greater intentions to assist, effortful behaviour from the perpetrator was found to incite anger from the HRP. A possible explanation for the unexpected finding is that, by responding to the feedback, the perpetrator accepts responsibility for their bullying actions and consequently HRPs feel anger towards the perpetrator for having caused the situation; that is, the perpetrator instigated the bullying situation. While not explored in this study, onset responsibility can affect perceptions of negative affect and helping intentions [41]. Outward demonstrations of anger from HRPs may hinder appropriate behaviour change by the perpetrator, thereby limiting the likelihood of future effortful behaviour.

Additionally, when targets of bullying did not make an overt (approach) confrontation toward the perpetrator, the HRP assigned less responsibility and felt more sympathetic toward the perpetrator. A possible reason for this discrepancy in HRP attributions could lie in an assumption that perpetrators have not been given the information by which to know that their behaviour needs to change, and are therefore less responsible for the ongoing situation. Central models of behaviour change (i.e., transtheoretical model, social cognitive theory, theory of planned behaviour) highlight the need for self-efficacy and cues to facilitate behavioural change [54]. It is worthy of note, and further investigation, that perpetrators were afforded leniency from HRPs for failing to make changes because of a (potential) lack of information regarding the situation, but targets who were frightened of or unaware of the procedures for dealing with workplace bullying were definitively less likely to receive help.

Furthermore, the demonstrated bias towards targets who utilise non-approach coping suggests a poor understanding, on the part of HRPs, of how workplace bullying plays out in organisations. A key feature of workplace bullying is a true and sustained power imbalance between the target and perpetrator, often manifesting as a supervisor/subordinate

relationship [55]. Expecting targets to approach and confront the perpetrator before escalating the situation to a HRP is problematic and is more likely to evoke further acts of bullying from the perpetrator than to resolve the situation.

*4.2. Practical Applications*

Data from this study suggests that HRPs may be biased in their judgement and attribution of responsibility (and subsequent helping responses) in cases of workplace bullying. Specifically, targets who displayed approach-coping behaviour were deemed less responsible and more worthy of help than targets who avoided any confrontation with the perpetrator. Such findings suggest that HRPs have a poor understanding of the nature of workplace bullying. Power imbalance is a key element of workplace bullying, often sustained through formal titles and roles and the acquisition and distribution of organisational resources; consequently, confronting and questioning perpetrators can result in serious and detrimental costs for the targets [56]. Research has consistently demonstrated that almost all preventive initiatives for workplace bullying are focused at the individual (target) level [57]; yet, rates of workplace bullying continue to climb [58]. Thus, although an obvious initiative lies in training employees to be more approach-orientated in situations of bullying, this is unlikely to minimise perpetrators' poor behaviour.

We posit that HRPs can play a significant role in a much-needed paradigm shift of workplace bullying mediation, intervention, and prevention, through better training in how workplace bullying plays out in organisations and how they may control their emotional responses. Specifically, HRPs would likely benefit from formalised training in recognising and responding to workplace bullying derived from academic research (which emphasises power imbalance between parties and highlights why targets may not engage in approach-coping behaviour). Additionally, HRPs could benefit from emotional regulation training. Typically delivered to healthcare workers in highly charged emotive settings [59] and individuals with mood disorders [60], emotional regulation training is designed to support mental health [61], detach emotionally from work [62,63], and adaptively manage emotions when necessary [64]. Recently, emotional regulation training has been expanded to generalised professional training programs, designed to 'strengthen or tune control processes that can support regulation to subsequently encountered events' [65] (p. 143). Successful trials have been documented in surgical [66], primary and secondary education [67], and consultancy settings [68]; thus, this training could be easily adapted to help HRPs recognise and limit their emotional responses, and examine situations of workplace bullying more impartially.

The vignette in this study also depicts a case of horizontal bullying (i.e., between two co-workers who hold varying levels of social power). Such cases are, anecdotally, less common than vertical bullying (i.e., between superiors and subordinates), as supervisors are explicitly powerful by virtue of their assigned title, access to resources, and ability to reward or punish their subordinates. Far from this detracting from the current findings, however, it is likely that tensions within the HRP role would be even more pronounced in cases of vertical bullying, especially where perpetrators formally or informally oversee the HRP themselves. Thus, HRPs may be even more likely to protect perpetrators and attribute blame and anger to targets in situations of vertical bullying. Accordingly, there is merit in the consistent presence of an alternative, independent 'champion' to protect the needs of employees, such as trade union representatives [69] or the ability to connect with a Fair Work Ombudsman [70]. Such parties are solely driven to accompany members to disciplinary or grievance hearings, engage in mediation with managers and supervisors to find resolutions to workplace issues, and work alongside WSH practitioners to develop best practice in workplace health and safety [71].

*4.3. Limitations and Future Directions*

Although our experimental model offers strengthened internal validity, the cross-sectional design limited our understanding of the sequencing of cognitions, emotional

responses, and behavioural intentions. Future iterations could evaluate the sequencing of the attribution–emotion model of stigmatization using a longitudinal design, for example. In addition, HRPs could start a diary when they are assigned a case of workplace bullying and complete the survey over multiple time frames (days, weeks) to see how their cognitions, emotions and behavioural intentions change; similar approaches have been used to understand how bullying causes harm to targets [72] and fluctuations in general wellbeing [73].

Secondly, the study focused on helping intentions relative to offset responsibility (i.e., who was responsible for the solution) to the exclusion of onset responsibility (i.e., responsibility for an origin of the problem) [41]. In practice, HRPs are likely to attribute offset responsibility partially based on the onset of the situation (i.e., who initiated the bullying event(s)). This is supported by our data which showed that HRPs considered both target and perpetrator behaviour when assessing perpetrators' responsibility for the situation. Future studies should thus evaluate HRPs evaluations of both onset and offset responsibility for both targets and perpetrators, to capture a more holistic view of HRPs decision making.

In this study, we position HRPs as responders to workplace bullying and measure their intended responses, based on a limited subset of information. In reality, HRPs are likely influenced by the broad responsibilities of their role [74], the culture of the organisation [75], and the severity of the bullying act [76], amongst other factors. Thus, future iterations may choose to add additional contextual factors (i.e., presence of a policy vs. no policy, perpetrator as a senior manager etc.) to the experimental design, to expand the full picture of HRP decision-making in the case of workplace bullying.

Finally, the study sample (*n* = 84) was not necessarily representative of the approximately 79,000 HRPs in Australia [77], although there was good diversity within the sample. Requests to advertise the study through various professional bodies were denied, perhaps signalling a reluctance to engage in this type of research (i.e., responding to workplace bullying). Emotional regulation training or workshops on advanced understandings of workplace bullying trialled under the banner of professional bodies would enable a more panoptic sample, and could inform principal training packages (i.e., university and other tertiary education programs).

## 5. Conclusions

Despite the considerable effort devoted to preventing the deleterious effects of workplace bullying, data suggests that it is an ongoing threat to individuals and organisations. Research has substantiated calls for psychosocial risk management from all levels of the organisation [57,72] and called on HRPs to play a key role in facilitating and overseeing specific risk management strategies [7]. Using the framework of the attribution–emotion model of stigmatisation, data from this study demonstrates a bias in attributions of responsibility and helping behaviour from HRPs, suggesting that they may hold outdated understandings of how workplace bullying manifests and/or feel significant pressure to protect the organisation at the expense of the individual (targets). This foundational knowledge in HRP helping behaviour warrants further scholarly inquiry and translation to practical outcomes.

**Author Contributions:** Conceptualization, A.M.N. and K.R.; methodology, K.R. and A.M.N.; formal analysis, K.R.; investigation, K.R. and A.M.N.; resources, A.M.N.; data curation, K.R.; writing—original draft preparation, K.R. and A.M.N.; writing—review and editing, A.M.N.; visualization, A.M.N.; supervision, A.M.N.; project administration, K.R. and A.M.N. All authors have read and agreed to the published version of the manuscript.

**Funding:** This research received no external funding.

**Institutional Review Board Statement:** The study was conducted according to the guidelines of the Declaration of Helsinki and approved by the Ethics Committee of the University of Queensland (2020001075).

**Informed Consent Statement:** Informed consent was obtained from all subjects involved in the study.

**Data Availability Statement:** Data can be made available upon request. Please address requests to the corresponding author.

**Conflicts of Interest:** The authors declare no conflict of interest.

## Appendix A

**Table A1.** Operationalisation of Coping Behaviours and Response Behaviours in the Experimental Vignettes.

| Workplace Bullying Scenario |
|---|
| Fred works in the operations department for a large utility company. Yesterday, Fred reported to the human resources representative that his co-worker Martin (who has been in the organisation much longer than Fred) had been belittling him in meetings and leaving him out of work assignments and social events with the team. Fred reported that the incidents started off small and then seemed to escalate over time. These behaviours have been going on for the last six months and have affected Fred's motivation to come to work. |

| Approach | Avoidance |
|---|---|
| One day, Fred confronts Martin and asks him to stop the remarks and behaviours. | One day, Fred tries to confront Martin about his behaviour but decides that he cannot do it. Instead, Fred moves away from Martin's work area. Fred discusses the situation with another co-worker who steps in and asks Martin to stop the remarks and behaviours that are negatively impacting Fred. |
| Effort | No Effort |
| Martin hears the feedback and resolves to start doing things differently, by reducing his belittling remarks and excluding behaviours toward Fred. | Martin hears the feedback but continues to behave in the same way, making negative remarks about Fred, and leaving him out of social activities. |

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
