# Peer review of "Human Resource Professionals’ Responses to Workplace Bullying"

_societies, doi:10.3390/soc12060190_

Round 1

Reviewer 1 Report

Overall, this is a very interesting and well written paper. The implications of workplace bullying for individuals and organisations are clearly laid out from the beginning. This provides a good rationale for the study which aims to deepen our understanding of HRPs responses to workplace bullying and the factors that shape such responses. This is important when considering how best to deal with / ameliorate the effects of workplace bullying. The attribution-emotion model of stigmatisation provides an appropriate theoretical underpinning for the study and the methodology is interesting given the potential difficulties of studying real life bullying situations. There are a few areas where more clarity is required to help improve the paper as suggested below: 

One of the key arguments early on in the paper is that HRPs find it difficult to deal with bullying given the competing tensions of their role - representing both management/leaders and employees. A major issue surrounding bullying in organisations is that the perpetrators are often management and bullying can be used as a means of employee control (Beale & Hoel, 2011) and this is referred to a few times in the paper. The opening sentence on page 1 refers to superiors or co-workers [L17] when defining bullying. However, the vignettes describe workplace bullying between co-workers where one party is inferior.

This raises two issues - (1) It wasn't clear what type of perpetrator-target relationship is being investigated - more details on what is meant by 'co-workers' in the vignettes could be provided as one party being 'inferior' to the other suggests a manager-subordinate relationship.

(2) There could be more explicit recognition at different points in the paper regarding how HRPs responses might differ if the bullying situation is vertical (between management and employee) or horizontal (between employees). It seems that the points raised in relation to the tensions within the HRP role would be far more pronounced when dealing with vertical bullying.

The link between research on bystander theory and HRPs is an interesting one. However, a key difference between typical bystanders and HRPs is that bystanders normally do not have an obligation (other than a moral one) to intervene whereas intervention / prevention of bullying is part of the role of the HRP – this difference could perhaps be recognised more upfront in the discussions on bystander intervention for example on page 2 and top of page 4. 

It could also be clarified what the role of the study participants was in relation to the vignettes they were provided with – i.e. were they being asked about their helping intention etc. in their role as a HRP rather than a bystander / colleague in the scenario? If participants were not specifically asked about their intentions in their role as HRP then this should be discussed – perhaps in the limitations.

It is interesting to note in the discussion on p10 the implications of the biases of HRPs towards targets (a sort of victim blaming approach). Again, here there is reference to the supervisor/subordinate relationship highlighting the points made earlier that more explicit recognition is required in relation to the type of bullying being discussed (vertical or horizontal / lateral). There could also be a link made here back to the point made early on in the paper about the contradictions within the HRPs role and their ‘loyalties’ to both management and employees. Perhaps an implication is that this highlights the need for an alternative, independent ‘champion’ to protect the needs of employees - such as trade union representatives. This is recognised in the final paragraph of the paper which suggests that HRPs feel significant pressure to protect the organisation at the expense of bullying targets. This contradiction in the HRPs role is a really interesting contribution of the paper and it would be good to see it explored in a little more depth throughout.

Minor points:

page 1 - provide more specifics and context around the 'industrial relations legislation' referred to (L43)

page 2 - 121/122 Should this read 'less care was provided for victims personally responsible for their situation i.e. the inebriated person?

Page 2 – 52/53 – clarify the change of context here from bullying to disciplinary scenarios e.g. have been know to demonstrate bias or stigmatization towards employees in other contexts …

page 4 - keep terms consistent in the hypotheses

page 4 (L194) - specify what is meant by 'recently' employed as HRPs

page 10 (L459) – delete 2nd thus

There are some minor spelling errors – thorough proofread needed.

Reviewer 2 Report

This is an interesting and good study. I have not much to comment on, but the following can be considered for further improvement of the paper:

(Line 17) Although workplace bullying is more commonly perpetrated by superiors/colleagues, it can be perpetrated by subordinates as well.

(Line 191-213) Inclusion and exclusion criteria not described.

(Line 203-204) What does preview tests mean and why were the 13 responses from preview tests removed? Needs clarification.

Statistical analyses not described under Methods section.

(Line 278-281) Missing the words “compared to” or “than”.

Table 1 and Table 2: Not clear what statistical test was used - suggest to insert as footnote in table. Same with Table 2.

All the best.
